# Interest in Rural Training Experiences in a Canadian Psychiatry Residency Program

**DOI:** 10.3390/ijerph192114512

**Published:** 2022-11-04

**Authors:** Jacquelyn Paquet, Vincent I. O. Agyapong, Pamela Brett-MacLean, Katharine Hibbard

**Affiliations:** 1Department of Psychiatry, Faculty of Medicine and Dentistry, University of Alberta, Edmonton, AB T6G 2B7, Canada; 2Department of Psychiatry, Faculty of Medicine, Dalhousie University, Halifax, NS B3H 4R2, Canada

**Keywords:** health professions education and training, rural and remote health, telehealth, information communication technology

## Abstract

Background: With the large majority of mental health professionals concentrated in urban settings, people living in rural and remote areas face significant barriers to accessing mental health care. Recognizing that early exposure is associated with future practice in rural and remote locations, we sought to obtain baseline data regarding interest in expanded rural residency training opportunities and academic teaching. Methods: In March 2021, all psychiatry residents at the University of Alberta (UofA) were invited to complete a 19-question survey that included both closed-ended (age, gender, year of study, rural experience, interest in rural training, etc.) and open-ended questions (challenges, barriers, academic training, and other comments). A reflexive thematic analysis using an inductive and semantic approach was completed on the comments. Results: 36 residents completed the survey (response rate, 75%). Significant associations were identified between previous rural training experience and interest in rural psychiatry training and practice. Female residents and junior residents were significantly more interested in rural training experiences than their counterparts. Thematic analysis noted concerns with the financial costs of accommodation and transportation, high service burden, continuity of care and isolation from their cohort. Many were interested in academic sessions on the realities of rural practice; approaches to collaborative care; and strategies on culturally relevant care; specifically Indigenous health. Conclusions: The University of Alberta has highlighted a focus on improving equity and accountability; and with a large rural catchment region; the residency program is well positioned to make training adjustments to diversify training. Based on our findings we have incorporated rural rotations for incoming residents and have developed further rural academic content to support our responsiveness and accountability to the rural and northern communities we are committed to serving. Future research should review the impact of rural training exposure in medical specialties on recruitment and retention as well as on healthcare outcomes.

## 1. Introduction

With over 9 million square miles, the second largest country in the world, Canada has a predominant rural landmass. Statistics Canada defines rural areas as having “less than 1000 people and a population density of fewer than 400 persons per square kilometer [1], while “rural populations in Canada are generally older, less affluent, and sicker” (p. 32), encompassing mixed rural/urban areas, rural areas close to cities, and those in more remote areas, the ethnosociocultural fabric of rural Canada is quite diverse. Many rural areas include large vibrant Indigenous communities that continue to experience systemic racism and ongoing struggles with intergenerational trauma and the legacy of residential schools, further predisposing members to mental illness [2,3]. Suicide rates in rural Alberta are at least three times higher across all age groups, with 31% of completed suicides occurring within the North zone, predominantly comprised of rural and remote communities, despite accounting for 10% of Alberta’s population [4]. Over two decades ago, the 2002 Romanow Report [5] identified the need to improve health care access in rural areas to uphold abiding principles of the Canada Health Act [6]. The pressing need has not abated, with the proportion of health professionals working in rural Canada markedly less than in urban settings. In 2020, Wilson et al. [7] noted 18% of Canadians live in rural communities, however only 8% of physicians practice rurally.

Fifty percent of cases of psychiatric illness present prior to age 14 and 75% before age 26, attesting to the chronicity of illness [8,9]. Early intervention is key to minimizing social and occupational consequences and yet access to psychiatric services is not widespread, especially in rural areas [8]. Rural communities are highly under-resourced in addiction and mental health support, including psychiatrists [10,11]. While comparisons are difficult given unique contexts and challenges in different countries, reduced access to mental health care services has also been reported in the United States (where 20% of the population lives in rural communities, served by 2% of specialists [12], and Australia (“major cities in Australia have approximately 15.1 full-time equivalent (FTE) employed psychiatrists per 100,000 population, while that figure was 5.8 for inner regional areas, 3.4 in outer regional areas” … “the more remote the location, the worse access is to psychiatric services” [13].

The Canadian Medical Association [14] reported Alberta’s ratio of psychiatrists per 100,000 population as 10.6, lower than the national average of 13.1, with most practicing in urban areas with rates as low as 3.2 in rural predominant provinces [3,15,16,17]. A review by the Alberta Physicians and Surgeons of Alberta in December 2021 revealed only 33 psychiatrists practice in rural communities and 53 in regional centers, comprising less than 15% of psychiatrists in Alberta yet serving approximately 30% of the province’s population [15]. Alberta has a centralized mental health system, with services of RAAPID (Referral, Access, Advice, Placement, Information and Destination) North and RAAPID South providing telephone consultations to rural and remote communities [18]. Should a patient be deemed acutely unwell, transfer to the nearest regional or urban center is arranged, and if deemed psychiatric, the psychiatric team may be consulted. However, the cost of transportation, long wait times for telepsychiatry assessment and removal from their social support system, all impact the feasibility, patient engagement and success of the service.

The literature on predictors of rural practice is quite inconsistent given variability in the rural exposure, including duration, voluntary versus mandatory and when in training it occurs [19,20]. Most studies, which have focused on rural family medicine practice, do note rural background as the strongest predictor; however this is not sustainable to meet the large discrepancy in rural care needs [20,21,22,23]. Even among those with rural interest, exposure to rural training increases their likelihood of engaging in rural practice and similarly, lacking rural exposure may reduce the likelihood of rural exposure [21]. Most note that early exposure is critical to increase rural practice, in a longitudinal and integrated experience to form meaningful interpersonal and community relationships [20,21,22,23]. Furthermore, pursuing family medicine or general practice is a strong predictor, with a lack of literature about specialty aims of rural practices despite the growing need [23,24]. For those who want to pursue rural practice, living in urban and metropolitan regions and often training in similar environments for medical school may dissuade rural practice, as they form relationships, acquiring urban role-models and meet a partner especially if their partner is highly educated [23,24]. Policies that may further promote a rural practice include loan repayments and other financial incentives to recruit and retain rural practitioners [24]. For generalists, rural communities may offer greater financial compensation, extended patient care needs and greater flexibility in their work environments [24]. Unfortunately, as noted above, research non-family medicine practice is absent, highlighting the need for further research-based programmatic initiatives to support comprehensive and collaborative rural care among other generalist specialties such as internal medicine, pediatrics, and psychiatry.

Alberta has prioritized equity of mental health outcomes, identifying underserved and marginalized groups, such as those living in rural or remote regions [25]. Alberta, a largely rural province, has just two metropolitan areas, Edmonton (north-central) and Calgary (south-central). Those residing in these areas comprise the majority of the province’s population. Health services in the province are organized into five geographic zones: North, Edmonton, Central, Calgary, and South. Rural medical training experiences increase the likelihood of rural practice, especially early in clinical training [26,27]. Despite a longstanding need, only recently have Canadian training programs begun to focus on rural psychiatry [2]—and much remains to be done.

Preparing psychiatrists for rural practice has been identified as an important strategy for addressing unmet mental health care needs in rural communities [8,16,17]. Interest in psychiatry has increased among graduating medical students. In 2021, 7.5% of medical students ranked psychiatry as their top choice specialty [28]. Each of the 10 provinces in Canada set limits on the family medicine and other specialty residency training positions they fund, including psychiatry. Rural-focused family medicine residency training has proved successful in improving numbers of rural family physicians [2,7] highlighting the importance of exposure to rural practice during residency, while also pointing to opportunities for integration and collaborative practice with community organizations and primary care providers.

Currently, the Psychiatry Residency Program at University of Alberta (U of A) is well-positioned to integrate different rural training pathways. With an increasing focus on social accountability in medicine, recognizing the need to address rural health inequities, the UofA Psychiatry Residency Program wanted to explore opportunities for expanding its rural training for residents.

There is little published literature on rural training programs in psychiatry nor academic frameworks to best prepare specialty residents for rural generalist practices. Prior to expanding our rural psychiatry training, we developed a survey to understand “trainee interests and concerns” (p. 4) as suggested in the Canadian Psychiatric Association’s position paper on training in rural and remote areas [29]. As psychiatry education leaders (as a department chair [VA], a residency program director [KH], a director of a residency program “learning environment” subcommittee [PB], and as a resident leader [JP, lead author], lived and trained in rural and remote communities), the authors share a strong motivation to diversify training to enhance marginalized and underserved communities’ access to mental health services.

The objectives of this study were to:(1)Identify the barriers to rural training and practice among psychiatry residents at the University of Alberta(2)Identify gaps in academic teaching related to rural related topics.(3)Determine strategies to initiate rural training opportunities for psychiatry residents at the University of Alberta.

## 2. Methods

### 2.1. Setting and Curriculum

This study was conducted in an academic center in an urban community in Canada. As a five-year program, the first year (PGY-1) focuses on basic clinical training with psychiatry-focused rotations organized through the remaining four years [29]. The program has up to ten residents per year with some attrition to other programs and delays due to personal circumstance. The program trains physicians with a broad experience in clinical and academic psychiatry. The first year (PGY1) is the basic training year with off service rotations, in 4-week intervals on complementary services to optimize their approach and differential to psychiatric diagnoses including rotations such as neurology, general internal medicine, family medicine and emergency medicine. This is followed by the second year (PGY2) of foundational psychiatry training in adult psychiatry in both in and outpatient settings, third year (PGY3) with a combination of inpatient, outpatient, and consultation services in subspecialty areas of child and adolescent, and geriatric psychiatry and the fourth year (PGY4) devoted to advanced psychiatry skills in consultative roles in consult liaison, rehabilitation, and primary care. Training is concluded with one year of elective time to consolidate their learning and determine their practice preferences. Rural exposure has been largely limited to seeing rural patients transferred to urban-based academic sites in Edmonton, Alberta, for assessments and admission, and telepsychiatry exposure during a 2-week community psychiatry rotation during their first year of the five-year program.

In 2020, the psychiatry programs nationally transitioned to competency by design, changing the focus from a time-based residency to a competency-based residency program [30]. Rather than a focus on years, there is a focus on stage of training, including introduction to discipline, comprising the first few months of residency, foundations of discipline, comprising the remainder of the PGY1 and PGY2, core of discipline for PGY3 and PGY4 and then transition to practice comprising most of PGY5 [30]. Within each stage there is a focus on learning objectives and competencies, which through regular observation and feedback is designed to promote skill acquisition for the specific specialty [30]. Within this framework there is a greater focus on marginalized populations and diverse practice environments to meet the needs of the population.

Residents have structured academic teaching one half day per week to align with their level of training and knowledge needs of their yearly rotations. Topics are focused on psychopharmacology, diagnoses, and psychotherapy to develop the clinical knowledge to become competent psychiatrists. Furthermore, residents attend weekly grand rounds presentations and quarterly journal clubs based on clinician interest and thus topics vary year to year depending on the area of expertise of the speaker. Speakers include resident psychiatrists, and both local, national, and international psychiatrists who have clinical expertise to promote continuing medical education of faculty, trainees, and scientists. The current academic curriculum hosts little in regard to rural practice. There is a cultural psychiatry day, although this is not focused on Indigenous health, nor rural issues. There are intermittent grand rounds discussions on Indigenous health, although this remains largely absent from residency training and continuing medical education.

Residents also complete psychotherapy training in diverse psychotherapy modalities, in individual and group settings to meet the royal college requirements and engage in weekly psychotherapy supervision. 

The residency program operates in publicly funded clinics and hospitals in the region and similarly residency positions are funded by the provincial government. The provincial government outlines the number of residents each specialty can train per year to address the expected healthcare demands [31]. Healthcare services are thus covered by provincial insurance, covering inpatient hospital, primary and specialty consults and follow up care and emergency department visits. Funding for healthcare is generated by federal and provincial taxes and provinces outline and deliver the care needed for their residents [31].

### 2.2. Survey and Analysis

In March 2021, all psychiatric residents at UofA were invited to complete a 19-question survey, including 9 short answer questions via RedCap to assess their baseline interest in expanded rural residency education (see Appendix A). All participants completed an online consent and data was anonymized. Exclusion criteria included not being an active resident. The survey took 10 min to complete. Participation was anonymous and no incentives were provided for participation. It remained open for eight weeks; a reminder email was sent after two and six weeks. The program director did not have access to the raw data. Following data collection, aggregate data were exported into MS Excel. Responses to closed-ended questions were analyzed to assess the influence of age, gender (male, female, nonbinary), relationship status (single, married, separated, divorced, other), and year of study (1st, 2nd, 3rd, 4th, 5th, other), as well as rural experience, interest in rural training and interest in rural practice. Quantitative data were analyzed using Student t-tests and chi-square tests. Qualitative analysis used an inductive and semantic approach with a reflexive thematic analysis of narrative responses ranging from a few words to a paragraph, exploring perceived barriers, and residents’ ideas for rural psychiatry training. Themes were identified with verbatim quotes selected as exemplars. The project was approved by the Health Research Ethics Board at the University of Alberta (Pro00106545).

## 3. Results

### 3.1. Sample Characteristics, Rural Experience, Knowledge, and Interest

In total, 36 of 48 residents completed the survey (response rate, 75%). Residents across all years responded: PGY-1, *n* = 8/10; PGY-2, *n* = 9/10; PGY-3, *n* = 6/10; PGY-4, *n* = 6/10; PGY-5, *n* = 7/8. More female residents (61.1%, *n* = 22/36) responded. The majority were married (55.5%, *n* = 20/36). Overall, a third (36.1%, *n* = 13/36) experienced rural practice for eight or more weeks during medical school, most of whom were female (61/1%; *n* = 8/12; *p* = 0.02) under 30 years of age (91.6%; *n* = 11/12) and junior residents at the time of the survey (*p* = 0.04).

Almost a third of respondents (30.5%, *n* = 11) were interested in enhanced rural psychiatry training, most of whom had been exposed to rural practice during undergraduate medical education (63%, *n* = 7/11; *p* = 0.012). More junior residents indicated interest in enhanced rural training (64%, *n* = 7/11).

Over 40% of residents (*n* = 15/36) responded that enhanced rural training experiences would maintain or improve their program ranking. Those who indicated it would worsen their program ranking were senior residents (67%, *n* = 8/12), whereas most residents who would maintain or improve their program ranking were junior residents (67%, *n* = 10/15). Those with rural exposure in medical school were more likely to respond that enhanced rural training would improve their ranking of the residency program (59%, *n* = 10/15; *p* = 0.01).

One-third of residents were interested in future rural practice (*n* = 12/36). Most residents who were not interested in rural practice had minimal rural exposure (<5 weeks; *n* = 7/10) whereas 50% of residents interested in future rural practice (*n* = 6/12) had 8 or more weeks of rural exposure (*n* = 6/10; *p* = 0.007). Few residents had knowledge of rural mental health supports or structures (19.4%, *n* = 7/36), although senior residents were more aware of available rural resources (71.4%, *n* = 5/7; *p* = 0.00008).

There was great interest in telepsychiatry training (94.4%, *n* = 34/36). Other popular options included combined telepsychiatry and intermittent clinics (66.7%, *n* = 24/36) and brief integrated blocks (41.7%, *n* = 15/36). Junior residents were more interested in rural training opportunities (*p* = 0.0002). Seven residents suggested additional options for enhancing rural psychiatry training: namely telepsychiatry only (*n* = 6) and longitudinal telepsychiatry (*n* = 1).

Percentages for sample characteristics and resident interest in different approaches to rural psychiatry exposure are presented in Table 1.

### 3.2. Thematic Analysis: Addressing Challenges Associated with Rural Psychiatry

Table 2 describes themes identified across narrative comments regarding (1) challenges and (2) opportunities resident respondents perceived in relation to rural training and practice. Many noted concerns regarding financial costs (transportation, accommodation) (*n* = 18), high service burden, limited support, and isolation from resident colleagues (*n* = 11). Other comments reflected concerns related to community integration, continuity of care, and rural conservatism. Regarding what they felt could help them prepare for rural psychiatry, respondents desired educational sessions on the realities of rural practice (*n* = 16) and culturally relevant care (*n* = 14). Additional comments addressed telepsychiatry teaching and collaborative care models.

## 4. Discussion

The University of Alberta Psychiatry Residency Program is well positioned to integrate rural training pathways given the support of leadership and the large rural communities the program serves [2,7,8,16,17,32]. Our survey provided valuable baseline data at a time when we are working to expand rural training. About a third of our respondents were interested in enhanced rural training—importantly, most were junior residents who are still exploring various career choices. From family medicine research, early exposure is influential on scope of practice, location, and career characteristics [2]. Successful programs such as University of Toronto, Australia, and New Zealand found that mandatory rural training and outreach exposure enhanced capacity and social accountability of the program [12,32,33]. Those interested in rural training had previous rural exposure in medical school, underlining the importance of rural exposure in undergraduate medical education similar to research on undergraduate medical education [21,22,23]. Further advocacy of rural training opportunities throughout medical training and promotion of rural medical schools or satellite sites may further optimize healthcare equity, providing primary care and specialty access more readily across Canada [17,19,20,21,22,23,24].

Multiple barriers were noted including separation from social support systems and increased workload and cultural differences that could impact resident integration and effectiveness. It is noteworthy that the survey was conducted one year into the COVID-19 pandemic, a context that may have served to highlight these concerns. Others have identified limited resources and support staff, increased administrative duties, limited opportunities for psychotherapy training, exam preparation, and continuing medical education as barriers to rural practice [12,17,34]. To best support residents engaging in rural training opportunities, training programs need to both prepare and support residents in rural settings [2,20]. Topics such as triage and interprofessional collaboration may reduce the barriers and help residents better integrate into the community [20,24] Given increased financial barriers of rural training, the rural health professions action plan (RhPAP), has provided funding for family medicine residents through subsidized or free accommodations to encourage and recruit practitioners to rural communities [35]. Similar opportunities have been provided to select specialties, however further advocacy of equitable access to opportunities for specialty and family medicine residents should be further optimized.

Given the process of selecting and recruiting residents in Canada, there are significant concerns regarding the attractiveness of residency programs. Benefits and consequences of rural and elective opportunities must be considered, and training requirements must be presented to incoming residents to attract and select residents in alignment with the mission, values and aims of the program [23]. Notably, while junior residents demonstrate more interest in rural training, some may view such requirements as burdensome and contrary to their career aims [24]. It is imperative to identify desires of residents, as mandatory rotations may affect a program’s ability to attract residents, whereas voluntary rotations may be unable to support the needed administrative structure [2,20,22]. A broad array of training sites should be considered, including rural sites near urban centers, regional centers, and rural and remote sites to meet the learning needs and objectives of trainees and their personal circumstances [24]. Moreover, diversity of training structures should be considered, including longitudinal experiences, brief integrated experiences and telepsychiatry to personalize training experiences and remain flexible given unique personal circumstances of the residents [22,24]. Almost all respondents were interested in formal telepsychiatry training. Other popular options included telepsychiatry combined with intermittent clinics and brief integrated blocks. Survey findings also highlighted clinical exposure combined with didactic teaching as providing an effective framework for a rural psychiatry curriculum.

### 4.1. Limitations

This survey focused on one training institution and despite a 75% response rate, only had 36 respondents. It is not known where the participants were raised, and although this is a known predictor of rural practice, it would have impacted the anonymity of the results. It would be imperative to repeat the survey in subsequent years to determine the effect of implemented changes on resident learning objectives and needs. Furthermore, the literature is absent in postgraduate specialty programs with most of the data from family medicine and undergraduate medical education. This can limit the generalizability to specialty programs, as most specialty training occurs in larger urban centers with fewer opportunities of engaging with rural communities.

### 4.2. Next Steps and Future Directions

The University of Alberta has begun to institute new clinical experiences for incoming junior residents including suburban, rural, and northern experiences with ongoing evaluations on the clinical experiences and teaching. A 2-week rural/suburban rotation has been introduced in PGY-1 in both Alberta and academic connections to the Yukon, a Northern arctic territory with a vast rural population. Virtual care has increased throughout the COVID-10 pandemic, facilitating greater integration into other rotations, and offering opportunities to better reach patients in rural areas. We view this as a critically important positive step in addressing rural mental health care inequities [4,22]. Although we have initiated a few sites, the number is expected to expand with more diverse community options. Furthermore, incoming residents can expand their rural experiences by completing their family medicine block in a rural community to further explore and integrate into communities. We are exploring other off-service opportunities as well as core psychiatry opportunities for PGY2 to PGY4 residents to engage longitudinally in rural communities to prepare residents to work with diverse and marginalized populations. Lastly, we are highlighting rural electives for our PGY4 and PGY5 residents to promote more autonomy, responsibility and skill development within rural communities, a critical period of determining their community of practice.

The U of A is reviewing its academic seminars across all training years to address challenges of wellness and burnout, interdisciplinary practice and optimizing our Indigenous lecture series to best prepare residents. Further equity, diversity, and inclusion initiatives to support marginalized population within the department of psychiatry. We continue to review and collaborate with other psychiatry programs nationally to inform our curricular developments and training opportunities and share our advancements with other programs. Parallel research includes reviewing provincial health data to identify gaps in service allocation in rural and remote communities and to determine the impact of reduced access to psychiatric care. We are collaborating with the Office of Rural and Regional Health to partner with rural family physicians to determine local needs, provide additional academic support and ultimately improve timely, accessible, and comprehensive care options. Given no to limited research on specialty practice in rural communities relevant to mental health, follow up surveys will be completed longitudinally to evaluate the impact of novel, place-specific (rural/remote) psychiatry training and academic curriculum on career choices and communities of practice.

## 5. Conclusions

Our resident survey represents an important starting point that has led to expanded rural curriculum and training options at the UofA. In partnership with others, including primary care physicians, we look forward to better addressing the mental health care needs of rural Albertans.

## Figures and Tables

**Table 1 ijerph-19-14512-t001:** Sample Characteristics, and Rural Experience, Interest and Knowledge.

	Total Respondents *n* = 36 (100%)	Options
Program Year	17/36 (47.2%)	Junior residents (PGY1,2)
	19/36 (52.8%)	Senior residents (PGY3, 4, 5)
Gender	22/36 (61.1%)	Female
	14/36 (38.9%)	Male
	0	Nonbinary
Relationship Status	9/36 (25%)	Single
	20/36 (55.6%)	Married
	0/36 (0%)	Separated/Divorced
	7/36 (19.4%)	Other
Age	2/36 (5.5%)	<25
	23/36 (63.9%)	26–30
	5/36 (13.9%)	31–35
	6/36 (16.7%)	>36
Experienced Rural Practice in Medical School	5/36 (13.9%)	0 weeks
	14/36 (38.9%)	1–4 weeks
	4/36 (11.1%)	5–8 weeks
	13/36 (36.1%)	>8 weeks
Interest in Rural Psychiatry Training	11/36 (30.5%)	Interested
	16/36 (44.4%)	Not interested
	9/36 (25%)	Unsure
Impact of Enhanced Rural Training on Residency Program Ranking	15/36 (41.7%)	Improve/Maintain
	9/36 (25%)	Unsure
	12/36 (33.3%)	Worsen
Aware of Rural Mental Health Support/Structures	7/36 (19.4%)	Yes
	16/36 (44.4%)	No
	13/36 (36.1%)	Unsure
Interest in Future Rural Practice	12/36 (33.3%)	Interested
	10/36 (28%)	Unsure
	10/36 (28%)	Not interested
	4/36 (10%)	Missing responses
Enhanced Rural Training Options	34/36 (94.4%)	Telepsychiatry
	10/36 (27.8%)	Intermittent clinics
	8/36 (22.2%)	Longitudinal clinics
	15/36 (41.7%)	Brief integrated blocks
	6/36 (16.7%)	Longer integrated blocks
	24/36 (66.7%)	Telepsychiatry & intermittent clinics

**Table 2 ijerph-19-14512-t002:** Resident perceived rural challenges and opportunity for preparation.

Theme	Subtheme	Description	Number of Respondents	Exemplar Quotes ^a^
Challenges of Rural Training	Financial	Out of pocket expenses related to travel and accommodation	*n* = 18/36 (50%)	*“Living in a rural location or having to travel without reimbursement for the time and cost of driving.”* (respondent 28) *“Issue with accommodations and housing. Plus difficulty for those with young families”* (respondent 30) *“It may force residents to have a car.”* (respondent 31)
	Practice Isolation	Finding strategies to manage the high care burden, limited collegial support and geographical separation from fellow residents	*n* = 11/36 (30.6%)	*“A lack of resources and lack of medical personnel.”* (respondent 5) *“Lack of supports, being in a small community and practicing something very sensitive.”* (respondent 10) *“Isolation from peers.”* (respondent 35)
Preparing for Rural Training	Navigating the reality of rural practice	Learning to triage, run a clinic and manage privacy. Learning how to approach common encounters, manage barriers, and the treatment of complex illness in rural settings	*n* = 16/36 (44.4%)	*“What to expect as not knowing is the scariest part.”* (respondent 11) *“Virtual tours of rural sites.”* (respondent 35) *“Having mental health workers from those communities talk about services they provide versus the need.”* (respondent 32)
	Providing culturally relevant care	Learning and practicing cultural safety with an awareness of intergenerational trauma and residential schools. Learning to identify and access mental health and addiction resources	*n* = 14/36 (38.9%)	*“Sessions with Indigenous community members on what is culturally safe.”* (respondent 7) *“Intergenerational trauma and residential schools”* (respondent 2) *“A list of mental health resources and services.”* (respondent 15)

^a^ Numbers indicate the source of the quoted text based on survey respondent’s unique identifier.

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
