# Peer review of "Interest in Rural Training Experiences in a Canadian Psychiatry Residency Program"

_ijerph, 2022, doi:10.3390/ijerph192114512_

Round 1
Reviewer 1 Report
General comments
1. This is important topic. A survey was done – in only 1 program – why go to all the trouble and not sample 5-10 programs? (IRB issues occur, but ‘should’ have a better sample.) This is myopic to Canada – is that what this International Journal wants? Readers are probably from other places.
There is a decent literature on rural training – many things outside of Canada are not covered. The manuscript
2. Presentation.
a. Readability: it is okay, but needs improvement in terms of detail/content and flow. Some of this also relates to basic manuscript formatting (e.g., Abstract should state relevance, current state and gap, then objectives).
b. Appendix does not really help us: a little information instead of the questions? Put all the questions in there.
c. Current international initiatives à Table. Columns for institution, authors/year, curricula, teaching methods, outcomes, and other factors for context.
Specific comments
Title.
1. Okay.
Abstract.
1. Need specifics here particularly in Methods:
a. Another line on type of thematic analysis and what/whose approach to it.
b. Add ‘anonymous’ if survey done so.
2. Conclusions see to presume to much; redo.
a. Need to collect data on broader sample.
b. Clarify needs/interests more specifically.
c. Future research should…
Introduction.
1. It is heuristically interesting, yet not very scholarly?
a. Define rural: many definitions across the world.
b. Predictors of rural practice?
c. Other things?
2. The basic approach…sort of missing:
a. Relevance.
b. Current state.
c. The gap (that the manuscript will fill).
d. Objectives.
Methods.
1. Need more for interested readers and researchers who may want to duplicate or build on. Consider paragraphs on…
a. Context/health system?
b. Design and basic curriculum: how much currently rural?
c. Participants: recruitment, consent, incentives and so on. They have had rural training? If so, what time, length, qualities of experience, curriculum? If not known à Limitation. Where did the come from (grow up rural)? If not known à Limitation.
d. Redcap is good. See also Eysenbach 2004 Impr Qual Web Surv Checklist in JMIR
e. Data collection/analysis: references for specific qual methods, please.
f. IRB? Redcap is secure and anonymous, but could folks still know who did and did not complete it based on age? Are the authors in leadership positions such that they could tell?
Results.
1. Okay but prose could be summarized better/more in a table?
2. See suggestions on tables.
Discussion.
1. Length and focus: consider into 4 paragraphs and make it more synthetic (and probably shorter, as very limited data in this small study).
a. Relevant findings.
b. Link with others’ findings.
c. Implications.
d. Limitations to add:
i. Only 1 residency.
ii. Small sample.
iii. No info given on whether grew up rural and other predictors of rural interest?
iv. Interest: yes, no or maybe? Likert would have been much better.
Conclusion.
1.
Tables/Figures
1. T1: odd formatting; hard to read; move first column to left.
2. T2: ditto; put in this arrangement…much easier to read.
|
|
|
|
|
|
References
1. Could be greatly improved.
Reviewer 2 Report
Thank you for the opportunity to review this paper on the important topic of improving rural communities’ access to mental health care, which is currently one of the most significantly under-resourced dimensions to rural health.
I offer the below feedback to assist the authors with further developing the manuscript.
Abstract
I would suggest re-considering the use of etc. in the abstract, it creates the impression the authors are vague about what they did and what they found, and there seems to be a couple of odd typos e.g. a question mark and dash before the word described line 20.
The line beginning ‘These findings have provided us with…’ is a little jarring because no context is provided first, the text goes straight from quite descriptive stats into how this is directing next steps, which doesn’t make sense. There is also no discussion of the findings from the analysis of open ended responses, just a short line to say thematic analysis was undertaken on these comments.
Introduction
There are a couple of abbreviations that would be helpful to explain for an international audience e.g. RAAPID.
In the last paragraph the authors begin talking about the University of Alberta’s Psychiatry Residency Program but half way through they use the terms ‘our’ and ‘we’ – I think it would be useful for the authors to explicitly address who they are and their roles in the context of this program.
Methods
Towards the end of the first paragraph the authors mention ‘qualitative analysis of narrative responses’ but in the opening of the paragraph where the survey is described there is no reference to the inclusion of open-ended questions just 19 questions overall – relatedly, if the survey took 10 minutes to complete, I don’t think the authors can claim they have ‘narrative responses’ as this suggests quite lengthy accounts. It is also unclear how these open ended responses were explored to develop themes (no specific description of qualitative methods of data analysis – the process of thematic analysis is not explained). More detailed information about the number of open-ended responses received and the depth to these responses would also be useful, e.g., there is a qualitative difference between a respondent taking the time to write a full paragraph or more in response to an open ended question and offering a single sentence.
Results
There is quite a lot of repetition in the way that the quantitative findings are presented, e.g., repeating under the age of 30.
Page 3 line 119 the authors say ‘other popular options’ but it is unclear what these options are for, other popular options for…
It should also be noted that the authors talk about gender but they have actually collected data on the binary sex respondents identified as (and it seems that not even an ‘other’ option was provided) rather than gender; this should be corrected throughout.
It is unclear what the term ‘narrative resident comments’ on p. 4 means.
Table 2 documents the name of the themes and sub-themes and offers an example extract that illustrates these, but it would be useful to have an actual description of the themes – currently the presentation of the analysis lacks depth.
Discussion
The first paragraph in this section reads like it really belongs in the introduction. Relatedly, more needs to be done in the discussion section to tell readers what the findings of this study suggest for future research and practice. For example, on p. 6 line 175 the authors mention the need for training programs to prepare and support residents in rural settings – what do the findings suggest in terms of how this should/can be better done? In the last paragraph on this same page there are a lot of references to what must be done, what is imperative and so on but these statements need to be contextualised and related directly to the findings of the study, and the work of others in this space.
Limitations and future directions
More detail is needed in terms of future directions/where to from here, and what this means for places behind the study site.
Conclusion and Next Steps
‘Next steps’ and ‘future directions’ seem to suggest the same thing, I would suggest combining and extending in one place, perhaps above and keep the conclusion simple.
The last paragraph of what is currently the conclusion reads again more like background for the introduction, but perhaps this is because the authors have not set up the content to be actions taken in response to the research? This needs clarification. – Re here Appendix A seems like content that should actually come up into the manuscript rather than supplementary material.
Round 2
Reviewer 1 Report
Good additions, edits and shifts in emphasis. Better Intro, Methods and Discussion.
Thank you.